# Reporting of outcomes in gastric cancer surgery trials: a systematic review

Bilal Alkhaffaf,[1,2,3] Jane M Blazeby,[4,5] Paula R Williamson,[6] Iain A Bruce,[7,8] Anne-Marie Glenny[9]

## ABSTRACT

**Background** The development of clinical guidelines for the surgical management of gastric cancer should be based on robust evidence from well-designed trials. Being able to reliably compare and combine the outcomes of these trials is a key factor in this process.

**Objectives** To examine variation in outcome reporting by surgical trials for gastric cancer and to identify outcomes for prioritisation in an international consensus study to develop a core outcome set in this field.

**Data sources** Systematic literature searches (Evidence Based Medicine, MEDLINE, EMBASE, CINAHL, ClinicalTrials. gov and WHO ICTRP) and a review of study protocols of randomised controlled trials, published between 1996 and 2016.

**Intervention** Therapeutic surgical interventions for gastric cancer. Outcomes were listed verbatim, categorised into groups (outcome themes) and examined for definitions and measurement instruments.

**Results** Of 1919 abstracts screened, 32 trials (9073 participants) were identified. A total of 749 outcomes were reported of which 96 (13%) were accompanied by an attempted definition. No single outcome was reported by all trials. 'Adverse events' was the most frequently reported 'outcome theme' in which 240 unique terms were described. 12 trials (38%) classified complications according to severity, with 5 (16%) using a formal classification system (Clavien-Dindo or Accordion scale). Of 27 trials which described 'short-term' mortality, 15 (47%) used one of five different definitions. 6 out of the 32 trials (19%) described 'patient-reported outcomes'.

**Conclusion** Reporting of outcomes in gastric cancer surgery trials is inconsistent. A consensus approach to develop a minimum set of well-defined, standardised outcomes to be used by all future trials examining therapeutic surgical interventions for gastric cancer is needed. This should consider the views of all key stakeholders, including patients.

**Correspondence to**
Bilal Alkhaffaf;
bilal.alkhaffaf@mft.nhs.uk

### Strengths and limitations of this study

► This systematic review is the first to describe the variation in outcome reporting within the field of surgical trials for gastric cancer.
► The study is based on a reproducible and transparent methodology which has been subjected to critical appraisal during a peer-review process.
► The study forms part of a larger project (the GASTROS study) to develop a 'core outcome set' for use in surgical trials for gastric cancer and was reviewed and funded by the National Institute of Health Research (UK).
► Only English-language and randomised studies were included in the analysis.
► Expanding the search may have resulted in the identification of other relevant outcomes reported in this field.

improve long-term survival, while minimising postoperative complications. Understanding which of these approaches are optimal for patients should be based on robust evidence from well-designed trials. This process involves the synthesis of evidence in the form of systematic reviews which can only be reliably undertaken if trials report the same outcomes and measure them in the same manner.

This review forms part of the first stage of a three-stage study, which intends to examine and address problems with inconsistent outcome reporting in gastric cancer surgery trials (GASTROS: GAstric Cancer Surgery TRials Reported Outcome Standardisation). The study aims to develop a 'core outcome set' (COS)—a minimum group of standardised and well-defined outcomes, relevant to key stakeholders and measured by all trials[3]—to standardise the reporting of outcomes in randomised control trials within this field. Our previously published study protocol contains an overview of all three stages.[4]

Within our study protocol, we described the results from a 'rapid review' of gastric cancer surgery trials during a 24-month period which demonstrated significant variations

## INTRODUCTION
### Background

Gastric cancer remains a leading cause of cancer-related death globally.[1] Long-term survival remains poor and has not improved significantly over the last four decades.[2] While there has been a shift to multimodal therapy over the last decade, surgery remains the primary method of curative treatment. Many developments in surgical techniques aim to

in outcome reporting. We hypothesised that these variations were likely to represent a more widespread problem within this field. Inconsistencies in outcome reporting are prevalent within the medical literature and contribute significantly to 'research waste'.[5] Several reviews have demonstrated that trials within the same field often report different outcomes, define them poorly and use various outcome measurement instruments.[6–9] This results in data which cannot be reliably compared or combined leading to further confusion within the evidence base. As such, initiatives such as Core Outcome Measures in Effectiveness Trials (COMET) were formed to promote the development of COS to address these issues.[3]

With respect to surgical trials for gastric cancer, (1) no rigorous examination of outcome reporting has been previously undertaken and (2) there is no COS for use in this field.

## Aims and objectives

This review aims to demonstrate whether further work to develop a COS to be used in surgical trials for gastric cancer is required. Specifically, the objectives are

1. To examine the degree of variation in the reporting of outcomes described by gastric cancer surgery trials.
2. To generate a 'long list' of potentially important outcomes which will be prioritised during a Delphi survey in stage 2 of the study.

## METHODS

### Definitions

The GASTROS study, and more specifically this review, focuses on outcome reporting in 'therapeutic surgical trials'. A 'surgical trial' has been previously defined as one of the following[10]:

► Type 1: A trial of medical interventions in surgical patients.
► Type 2: A trial which compares a surgical intervention to another surgical intervention.
► Type 3: A trial which compares a surgical intervention to a non-surgical intervention.

The GASTROS study focuses on 'type 2' trials due to the significant research activity within this field (a detailed justification can be found in our study protocol).[4] In the context of gastric cancer, a 'therapeutic surgical intervention' is defined as a potentially curative procedure which aims to excise the gastric neoplasm resulting in partial or total organ loss.

### Search strategy

A summary of the review's inclusion and exclusion criteria is summarised in table 1, with details of our search strategy presented below. An example search algorithm for the MEDLINE via OVID database is presented in supplementary appendix 1.

### Timeline

Trials were searched from 1996 when the first Consolidated Standards of Reporting Trials statement for the reporting of randomised controlled trials (RCTs) was published up to and including March 2016. The COS aims to influence trial design regardless of the country of origin of participating centres and patients. Many of the surgical interventions (eg, D2 lymphadenectomy) which have recently been examined in the West have long been established practice in the Far East and Asia. Searching trials over a 20-year period allows a comprehensive understanding of which outcomes have been measured for similar trials regardless of the trial location.

| Table 1 | Inclusion and exclusion criteria for this review | |
|---|---|---|
| | **Included** | **Excluded** |
| Types of studies | ► Type 2* surgical randomised controlled trials (RCTs) and protocols of surgical RCTs (all trial phases).<br>► Systematic reviews of type 2 surgical RCTs.<br>► English-language studies | ► Type 1 or type 3* surgical RCTs and systematic reviews of type 1 or type 3 RCTs<br>► Non-randomised studies<br>► Non-English-language studies |
| Population | ► Patients aged 18 years and over | ► Patients below the age of 18 |
| Interventions | ► Partial or total gastrectomy<br>► Surgery with curative intent | ► Oesophagectomy for gastro-oesophageal junctional tumours<br>► Surgery with non-curative intent (ie, in stage 4 cancer with prior expectation of an R1 or R2 resection) for the relief of symptoms such as gastric outlet obstruction or bleeding<br>► Endoscopic interventions |
| Conditions | ► Invasive cancer of the stomach and gastro-oesophageal junction | ► Dysplasia or non-invasive gastric neoplasms.<br>► Sarcoma (including gastrointestinal stromal tumours)<br>► Gastric lymphoma |

*Type 1: a trial of medical interventions in surgical patients; type 2: a trial which compares a surgical intervention to another surgical intervention; type 3: a trial which compares a surgical intervention to a non-surgical intervention.[10]

## Identifying studies

Detailed search strategies were developed for each of the following electronic databases examined:

► Evidence Based Medicine Reviews via OVID
  – Cochrane Database of Systematic Reviews 2005 to 30 March 2016.
  – ACP Journal Club 1991 to March 2016.
  – Database of Abstracts of Reviews of Effects 1st Quarter 2016.
  – Cochrane Central Register of Controlled Trials February 2016.
  – Health Technology Assessment 1st Quarter 2016.
  – NHS Economic Evaluation Database 1st Quarter 2016.
► MEDLINE via OVID (1 January 1996 to 30 March 2016).
► EMBASE via OVID (1 January 1996 to 30 March 2016).
► CINAHL via EBSCO (1 January 1996 to 30 March 2016)

In order to identify surgical interventions and outcome measures being used in current studies, we searched the following databases for protocols of ongoing trials, including completed trials not yet published:

► The US National Institutes of Health Trials Register (http://clinicaltrials.gov).
► The WHO International Clinical Trials Registry Platform (http://apps.who.int/trialsearch/default.aspx).

Non-English-language studies were excluded from this review due to resource limitations. Trials published only as conference abstracts were excluded as they are often limited by 'word count' and hence the abstract would not represent a comprehensive list of outcomes measured in the respective study.

## Assessment of eligibility

For quality assurance, two review authors (BA and AMG) independently screened the titles and abstracts retrieved from the electronic searches. This assessment was undertaken in groups of ten abstracts in reverse chronological order. Once there was complete agreement with two consecutive groups of 10 abstracts, the remaining abstracts were split and each reviewer screened independently. Full-text copies of all study publications that appeared to meet the inclusion criteria were obtained. Full-text copies were also obtained where there was insufficient information in the title or abstract to make a clear judgement. Systematic reviews of RCTs were also retrieved to find studies which had previously not been identified.

BA and AMG independently assessed the full-text copies for eligibility. This assessment was undertaken in groups of 10 publications in reverse chronological order. Once there was complete agreement with two consecutive groups of 10 abstracts, the remaining publications were split, and each reviewer extracted data independently. Any disagreements were resolved through discussion. There were no unresolved disagreements that required referral to the GASTROS study management team for a final decision.

## Data extraction

BA and AMG independently reviewed all eligible publications and extracted data (described below) into a Microsoft Excel (V.2013, Microsoft, Washington, DC, USA) spreadsheet.

## Publication versus study

It is not uncommon that investigators publish results at different stages of their trial and with each publication present a new set of outcomes. The GASTROS study team decided to amalgamate the outcomes published in all publications associated with a single trial to more fairly reflect outcomes being reported by research groups.

## Trial characteristics

The following data were recorded for each trial:
1. Author details.
2. Title of publication.
3. Journal.
4. Year of publication.
5. Number of participating centres.
6. Country of first author.
7. Countries of participating centres.
8. Total number of patients recruited to the study.
9. Length of follow-up.
10. Interventions being investigated.

## Outcomes

We defined an outcome as 'a unique endpoint which attempts to describe health-related changes that occur secondary to a therapeutic intervention'.[4] The following data were recorded for each outcome:

1. Outcome measured (and whether stated as primary or secondary outcome). Where a primary outcome was not explicitly stated, the outcome on which the sample size calculation was based was taken as the primary outcome.
2. Whether the outcome was defined or not. Outcomes were considered defined if text of their meaning or a citation was provided.
3. The definition of the outcome.
4. The method of outcome measurement (indicators and/or tools used, if relevant).
5. Time points and time period at or during which the outcome was measured (eg, quality of life at 3 months post surgery).

## Merging outcomes and grouping under 'themes'

Outcomes were extracted verbatim from publications and minimal merging of terms was undertaken. Outcomes were merged to accommodate for variant spellings of the same words. For example, 'anastomotic leak', 'anastomotic leak**age**' and 'anastomotic leak**s**' were merged into 'anastomotic leak'. The verbatim texts and merged terms were verified and authorised respectively by the study management group.

From the experience of other groups undertaking reviews of outcome reporting, the resulting lists of outcomes are generally extremely long and unwieldy.[6]

Table 2  Number of times at least one outcome from respective theme was reported in published trials and unpublished or actively recruiting trial protocols

| Outcome theme | Theme definition | Published trials (n=32) (%) | Unpublished or actively recruiting trial protocols (n=23†) (%) |
|---|---|---|---|
| Cost | Relating to delivery of surgery as part of clinical care within a healthcare system | 1 (3) | 5 (23) |
| Patient pathway | Outcomes related to the flow of patients through the healthcare system (eg, hospital stay, readmission) | 20 (63) | 4 (17) |
| Patient-reported outcomes | Outcomes taken from the patient perspective* | 6 (19) | 13 (57) |
| Surviving and controlling cancer | Measures of disease recurrence or disease progression | 15 (47) | 17 (74) |
| Mortality | Outcomes related to short-term and long-term survival/death rates and cause of death | 27 (84) | 19 (83) |
| | Short-term mortality/perioperative death | 27 (84) | 10 (43) |
| | Long-term survival | 13 (41) | 19 (44) |
| Technical aspects of surgery | Outcomes recorded directly in the operating theatre (eg, operation time, blood loss) | 31 (97) | 13 (57) |
| Recovery from surgery | Report of patient condition following surgery and the ability to return to preoperative or premorbid state | 16 (50) | 9 (39) |
| Adverse events | Forms of short-term and long-term postoperative complications following surgery | 31 (97) | 18 (78) |

*Certain patient-reported outcomes may fall under other 'themes', for example, 'postoperative pain' may relate to 'recovery from surgery'.

†One trial protocol contained no information about planned outcomes to report, therefore 23 out of total 24 trials were included in this table.

Consequently, developing a method to organise these outcomes has been necessary. The subject of taxonomy in outcome reporting, including hierarchical structure and which terms/definitions to use, is an emerging area of great significance. We set out our definitions a priori, which can be found in our study protocol.[4] Many COS developers have organised their outcomes into broad categories with common 'themes'. Our study is one of only a handful addressing outcome reporting in surgical trials related to the gastrointestinal (GI) tract. At the time of data analysis, we opted to group outcomes under 'themes' (detailed in table 2) similar to those described by other surgical COS.[6–8 11] Doing so enables COS researchers to more readily understand trends in outcome reporting within the field of GI surgery. While the themes used in our review enable the reader to understand the types of outcomes being reported, this system has not been developed through wider consensus and has not been subject to a validation process.

At the time of writing, a broader taxonomy for outcome classification had been proposed.[12] This system aims to address some of the ambiguity associated with outcome classification on a wider-scale and organises outcomes under 38 'outcomes domains' which sit under five 'outcomes areas' ('mortality/survival', 'physiological/ clinical', 'life impact', 'resource use' and 'adverse events'). While the authors have demonstrated that this system is comprehensive and applicable to trials irrespective of the field being studied, they have called for further validation of their work.

### Patient and public involvement

A Study Advisory Group (SAG) forms part of the management structure of the wider GASTROS study,[4] of which this review forms part of the first stage. The SAG is made up of key stakeholder representatives including patients, oncology nurses and surgeons. The group provides advice on the methodology of the study, general delivery of the study against its stated objectives and ensures that the viewpoints of all stakeholder groups are considered. The results of this systematic review were presented to a SAG meeting; the ensuing discussion influenced certain aspects of the results section within this paper such as the emphasis on patient-reported outcomes.

### RESULTS
### Summary

A total of 1919 abstracts were screened which resulted in the identification of 48 publications from 32 trials (having recruited a total of 9073 patients) eligible for inclusion in the review (figure 1). A further 875 protocols were screened which identified 24 active or unpublished trials aiming to recruit 10 761 patients. A summary of all trials included in the analysis is described in table 3. During the data extraction process, no disagreements requiring

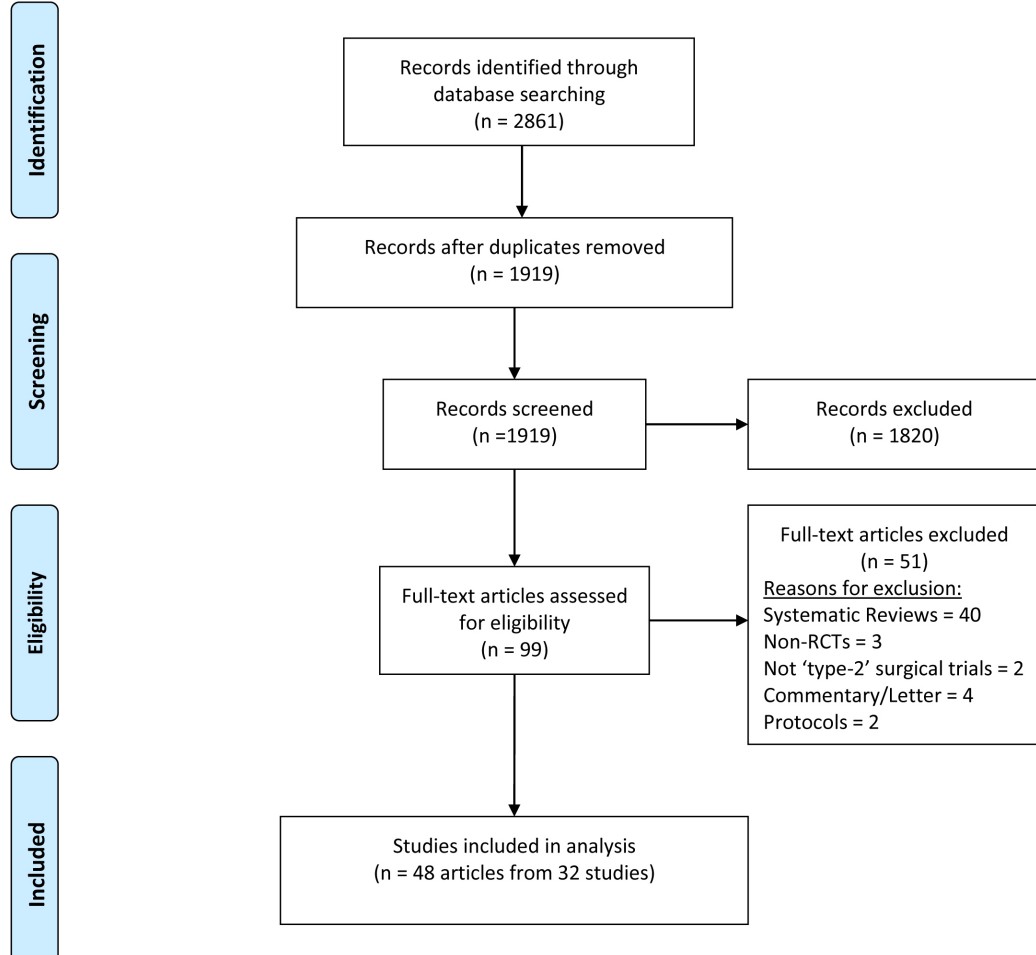

**Identification**

Records identified through database searching
(n = 2861)

**Screening**

Records after duplicates removed
(n = 1919)

Records screened
(n =1919)

Records excluded
(n = 1820)

**Eligibility**

Full-text articles assessed for eligibility
(n = 99)

Full-text articles excluded
(n = 51)
Reasons for exclusion:
Systematic Reviews = 40
Non-RCTs = 3
Not 'type-2' surgical trials = 2
Commentary/Letter = 4
Protocols = 2

**Included**

Studies included in analysis
(n = 48 articles from 32 studies)

**Figure 1** Study selection and inclusion.

discussion in the study management group arose between the two independent reviewers. A total of 749 (392 unique terms) outcomes were reported of which 13% (96 out of 749) were accompanied by an attempted definition. Thirty-eight per cent of trials (12 out of 32) described a primary outcome or provided a sample size calculation. No single outcome was reported in every trial.

### Analysis of outcomes according to themes

Outcomes were organised into eight 'outcome themes', illustrated in figure 2 and described in table 2. A comprehensive list of reported outcomes is presented in supplementary appendix 2. Below, we present a summary of some of the most commonly reported short-term and long-term outcome themes.

### Mortality

Death after surgery was generally described as 'short-term' and 'long-term' survival. Long-term survival was used as a primary outcome measure in 41% of trials (13 out of 32). The terms used to describe long-term mortality and the time points at which they were measured was inconsistent (table 4). 'Short-term' mortality was reported by 84% of trials (27 out of 32) of which 15 provided one of the

following definitions (frequency each definition was used is presented in brackets).
1. 'Death within 30 days of surgery' (3/15).
2. 'Death of any cause within 30 days, or death within the same hospitalisation' (9/15).
3. 'In-hospital deaths and 'deaths' within 1 month' (1/15).
4. 'Death within 30 days of the operation or during any hospital stay' (1/15).
5. 'Any death that occurred during the hospital stay' (1/15).

### Adverse events

Adverse events were the most common outcome theme to be reported and made up half of the 10 most reported outcomes (table 5). 'Anastomotic leak' was the most common adverse event to be reported and was described using five different definitions (frequency each definition was used is presented in brackets).
– 'Clinical and radiological diagnosis' (2).
– 'Confirmed by gastrointestinal x-ray imaging, endoscopy, or angiography' (1).
– 'Dehiscence confirmed by radiographic examination using contrast medium' (1).

**Table 3** Published gastric cancer surgery trials included in the study analysis

| Trial reference(s) | Trial number | Author | Year of first publication | Participating countries | Recruiting sites (n) | Patients recruited | Interventions |
|---|---|---|---|---|---|---|---|
| 39–42 | 1 | Hartgrink et al, Bonenkamp et al, Sasako et al, Songun et al | 1995 | Netherlands | 80 | 1078 | ▲ D1 lymphadenectomy<br>▲ D2 lymphadenectomy |
| 43 44 | 2 | Cuschieri et al | 1996 | UK | 32 surgeons* | 737 | ▲ D1 lymphadenectomy<br>▲ D2 lymphadenectomy |
| 45 46 | 3 | Marubini et al, Bozzetti et al | 1999 | Italy | 31 | 615 | ▲ D2 subtotal gastrectomy<br>▲ D2 total gastrectomy |
| 47 | 4 | Maeta et al | 1999 | Japan | 1 | 70 | ▲ D3 total gastrectomy<br>▲ D4 total gastrectomy |
| 48 | 5 | Furukawa et al | 2000 | Japan | 1 | 110 | ▲ Total gastrectomy and distal pancreatectomy<br>▲ Pancreas preserving total gastrectomy |
| 49 | 6 | Csendes et al | 2002 | Chile | 1 | 187 | ▲ D2 total gastrectomy with splenectomy<br>▲ Spleen preserving D2 total gastrectomy |
| 50 | 7 | Kitano et al | 2002 | Japan | 1 | 28 | ▲ Laparoscopic assisted distal gastrectomy<br>▲ Open distal gastrectomy |
| 51 | 8 | Fujii et al | 2003 | Japan | 1 | 20 | ▲ Laparoscopic assisted distal gastrectomy<br>▲ Open distal gastrectomy |
| 52–54 | 9 | Degiuili et al | 2004 | Italy | 5 | 267 | ▲ D1 lymphadenectomy<br>▲ D2 lymphadenectomy |
| 55–58 | 10 | Sano et al., Sasako et al, Kodera et al, Tsujinaka et al | 2004 | Japan | 24 | 523 | ▲ D2 lymphadenectomy<br>▲ D2 lymphadenectomy and para-aortic node dissection |
| 59 | 11 | Shibata et al | 2004 | Japan | 9 | 81 | ▲ Pylorus-preserving gastrectomy<br>▲ Distal gastrectomy |
| 60 | 12 | Inaba et al | 2004 | Japan | 1 | 410 | ▲ Midline laparotomy<br>▲ Transverse laparotomy |
| 61 | 13 | Lee et al | 2005 | South Korea | 1 | 47 | ▲ Laparoscopic-assisted D2 distal gastrectomy<br>▲ Open D2 distal gastrectomy |
| 62 | 14 | Hayashi et al | 2005 | Japan | 1 | 28 | ▲ Laparoscopic assisted distal gastrectomy with extra-perigastric node dissection<br>▲ Open distal gastrectomy with extra-perigastric node dissection |
| 63 | 15 | Huscher et al | 2005 | Italy | 1 | 59 | ▲ Laparoscopic subtotal astrectomy<br>▲ Open subtotal gastrectomy |
| 64–67 | 16 | Wu et al | 2006 | Taiwan | 1 | 221 | ▲ D1 lymphadenectomy<br>▲ D3 lymphadenectomy |

Continued

**Table 3** Continued

| Trial reference(s) | Trial number | Author | Year of first publication | Participating countries | Recruiting sites (n) | Patients recruited | Interventions |
|---|---|---|---|---|---|---|---|
| 68 69 | 17 | Sasako et al Kurokawa et al | 2006 | Japan | 27 | 167 | ▲ Left Thoracoabdominal approach<br>▲ Abdominal trans-hiatal approach |
| 70 | 18 | Yu et al | 2006 | South Korea | 1 | 207 | ▲ Total gastrectomy with splenectomy<br>▲ Spleen preserving total gastrectomy |
| 71 | 19 | Kulig et al | 2007 | Poland | 6 | 275 | ▲ D2 lymphadenectomy<br>▲ D2 lymphadenectomy and para-aortic node dissection |
| 72 73 | 20 | Kim et al | 2008 | South Korea | 1 | 164 | ▲ Laparoscopic assisted distal gastrectomy<br>▲ Open distal gastrectomy |
| 74 | 21 | Yonemura et al | 2008 | Japan, Taiwan, South Korea | 10 | 269 | ▲ D2 lymphadenectomy<br>▲ D4 lymphadenectomy |
| 75 76 | 22 | Kim et al | 2010 | South Korea | 13 | 1416 | ▲ Laparoscopic assisted D2 distal gastrectomy<br>▲ Open distal D2 gastrectomy |
| 77 | 23 | Imamura et al | 2011 | Japan | 11 | 210 | ▲ D2 gastrectomy and bursectomy<br>▲ D2 gastrectomy without bursectomy |
| 78 | 24 | Cai et al | 2012 | China | 1 | 123 | ▲ Laparoscopic assisted distal gastrectomy<br>▲ Open distal gastrectomy |
| 79 | 25 | Chen Hu et al | 2012 | China | 1 | 88 | ▲ Laparosopic gastrectomy<br>▲ Open gastrectomy<br>▲ Standard postoperative protocol<br>▲ Fast-track postoperative protocol |
| 80 | 26 | Takiguchi | 2013 | Japan | 1 | 40 | ▲ Laparoscopic assisted distal gastrectomy<br>▲ Open distal gastrectomy |
| 81 | 27 | Lee et al | 2013 | South Korea | 1 | 204 | ▲ D2 distal gastrectomy<br>▲ D2 total gastrectomy |
| 82 | 28 | Sakuramoto et al | 2013 | Japan | 1 | 64 | ▲ Laparoscopic assisted distal gastrectomy<br>▲ Open distal gastrectomy |
| 83 | 29 | Aoyama et al | 2014 | Japan | 1 | 26 | ▲ Laparoscopic assisted distal gastrectomy<br>▲ Open distal gastrectomy |
| 84 | 30 | Hirao et al | 2015 | Japan | 11 | 210 | ▲ D2 gastrectomy with bursectomy<br>▲ D2 gastrectomy without bursectomy |
| 85 | 31 | Galizia et al | 2015 | Italy | 1 | 73 | ▲ D1 '+' total gastrectomy<br>▲ D2 total gastrectomy |
| 86 | 32 | Hu et al | 2016 | China | 14 | 1056 | ▲ Laparoscopic assisted D2 distal gastrectomy<br>▲ Open D2 distal gastrectomy |

*Number of sites not specified.

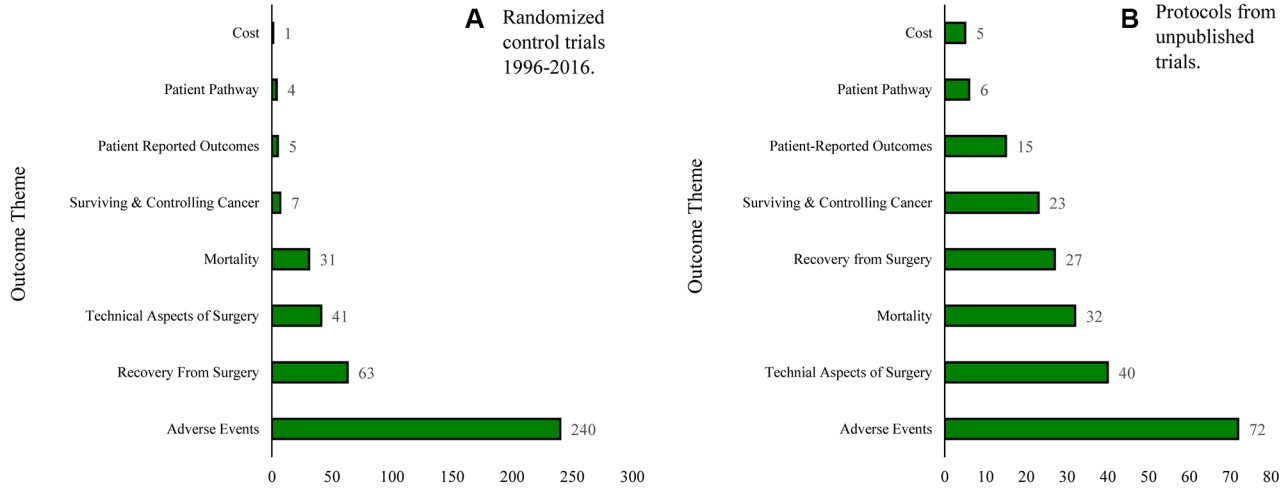

**Figure 2** Outcome themes reported in (A) gastric cancer surgery trials and (B) in future trials based on study protocols.

– 'Type I anastomotic leakage: a small localised leakage at the oesophagojejunal anastomosis, without pleural or abdominal spillage, demonstrated by radiologic studies with barium' (1).

– 'Type II anastomotic leakage: an important dehiscence of the oesophago-jejunal anastomosis, with pleural or abdominal dissemination, appearance of intestinal content through the drains, a positive methylene blue test (appearance of orally ingested methylene blue through the drains), and clear demonstration of this leakage by radiologic contrast studies' (1).

Adverse events were categorised by 12 out of 32 trials using terms including 'major', 'minor', 'short-term' and 'long-term'. Five trials used a formal classification system (Clavien-Dindo Classification or Accordian Severity Grading).

**Table 4** Reporting of 'long-term mortality' in gastric cancer surgery trials

| Term used | Trials reporting outcome (n) | Follow-up period used | Frequency defined | Definitions provided |
|---|---|---|---|---|
| 'Overall survival' including 'death from all causes' | 19 | Not described<br>3 years<br>5 years<br>6 years<br>7 years<br>10 years | 7 | ▶ Date of randomisation until the day of death or the day of last follow-up (censored)<br>▶ Date of surgery to the date of death from any cause, censoring the follow-up time at the most recent date for living patients<br>▶ Date of randomisation to the date of death<br>▶ Date of randomisation to the date of death from any cause<br>▶ Overall survival included operative deaths<br>▶ Overall survival excluded postoperative deaths |
| 'Survival' including 'survival period' | 13 | Not described<br>5 years<br>11 years | 4 | ▶ Survival excluding operative mortality<br>▶ Survival 5 years after curative surgery |
| 'Disease-specific survival' including 'gastric cancer-related deaths' | 4 | Not described<br>5 years | 1 | ▶ Proportion of patients who had not died from gastric cancer |
| Disease-free survival* | 1 | Not described | 1 | ▶ Time from randomisation to recurrence or death due to any cause |
| Recurrence-free survival* | 2 | Not described | 2 | ▶ Time from randomisation to either the first recurrence or death from any cause<br>▶ Time from randomisation to the first documentation of cancer recurrence or death from any cause |

*Although these terms do not relate to mortality, they have been included in this table as the definitions provided by papers describe death as an end point.

**Table 5** The 10 most frequently reported outcomes

| Outcome | Theme | Trials reporting outcome (n) | Trials reporting the outcome* |
|---|---|---|---|
| Number of lymph nodes dissected/resected/retrieved | Technical aspects of surgery | 22 | 2, 3, 5, 7, 9, 10, 13, 14, 15, 16, 17, 18, 19, 20, 21, 23, 24, 25, 26, 29, 31, 32 |
| Operative time | Technical aspects of surgery | 18 | 4, 5, 7, 8, 10, 11, 13, 14, 15, 16, 20, 22, 23, 24, 25, 26, 27, 30 |
| Anastomotic leak | Adverse events | 17 | 1, 2, 4, 6, 10, 14, 16, 17, 19, 22, 23, 24, 25, 27, 28, 31, 32 |
| Pancreatic fistula | Adverse events | 15 | 1, 2, 4, 5, 10, 16, 17, 19, 22, 23, 26, 27, 28, 29, 32 |
| Duration of hospital stay | Patient pathway | 12 | 1, 2, 5, 6, 7, 9, 13, 15, 20, 23, 26, 28 |
| Duration of postoperative hospital stay | Recovery from surgery | 11 | 2, 7, 9, 11, 14, 16, 19, 24, 25, 27, 32 |
| Pneumonia | Adverse events | 11 | 7, 10, 11, 12, 17, 23, 27, 28, 29, 31, 32 |
| '5-year' survival | Mortality | 11 | 1, 2, 6, 9, 10, 16, 17, 18, 19, 22, 26 |
| Wound infection | Adverse events | 10 | 2, 6, 12, 15, 16, 18, 19, 22, 24, 28 |
| Abdominal abscess | Adverse events | 10 | 5, 9, 10, 16, 17, 19, 23, 26, 28, 29 |

*See table 1 for trial numbers and associated publications.

### Patient-reported outcomes

Patient-reported outcomes (PROs) were reported in 19% of trials (6 out of 32) and included measures of quality of life (QoL) (n=3) and 'pain' (n=3). QoL was measured using validated tools for gastric cancer in two trials (EORTC QLQ-C30 with QLQ and Spitzer QoL Index) and a non-validated tool in one trial. Pain was measured using three different visual-analogue scales.

### Multicentre trials

Forty per cent (13 out of 32) of studies were multicentre trials (table 2). Ninety-two per cent (12 out of 13) of trials stated a primary outcome measure (5-year overall survival, 5-year survival, all-cause mortality and 3-year disease-free survival). Sixteen per cent (63 out of 393) of all outcomes reported in these trials were accompanied by an attempted definition. PROs were reported in 23% of trials (3 out of 13) with 8% (1 out of 13) of studies reporting 'QoL' as an outcome.

### Findings from study protocols

Most of the 24 ongoing or unpublished trials[13–36] are recruiting in China (n=13), with 20 examining 'extent of lymphadenectomy' or minimally invasive approaches to surgery. A total of 220 uniquely termed outcomes are planned to be reported, 35 of which (16%) have an accompanying definition in the respective protocol. The most common term used to report 'long-term survival' is 'overall survival' (OS) which will be measured by 16 trials. Seven of these trials plan to measure OS after 5 years of follow-up, three at 3 years of follow-up and six did not identify time points at which OS would be measured. At the time of our search, one trial protocol contained no information about which outcomes are to be measured.

QoL is due to be measured by 10 trials (42%) with five trials proposing to use one or a combination of four different measurement instruments (EORTC QLQ-C30 with QLQ-STO22, SF-36, GIQLI and Euro-Quality of Life-5D). Seven protocols described the timing of the quality of life measurements as follows:

► 'Preoperative, postoperative 3 weeks and postoperative 12 months'.
► 'In pre-therapy <7 days, preoperative <7 days and postoperative at 12 months after surgery'.
► '90th postoperative day'.
► 'Regularly for 3 years after surgery'.
► 'Preoperatively, 5 days postoperatively, 3 months, 6 months and 1 year postoperatively'.
► 'Baseline, 1 week, 1 month, 6 month, 1 year, 3 years'.
► '6 weeks, 12, 24, 36, 48 and 50 months after surgery'.

### DISCUSSION

This review is the first to examine the subject of outcome reporting in this field and demonstrates significant inconsistencies in the outcomes measured in trials examining therapeutic surgical interventions for gastric cancer. Not only is there disagreement about 'what' outcomes should be measured, but also 'when' and 'how' they should be measured. Consequently, undertaking meta-analyses and systematic reviews of these interventions becomes problematic and impacts negatively on the ability of researchers and clinicians to formulate robust clinical guidelines for the radical treatment of gastric cancer.

This problem is not confined to previously reported trials. Our analysis of outcome reporting in multicentre studies and review of active trial protocols has demonstrated similar issues and further highlights the potential 'research waste' within this field. Glasziou and Chambers

estimate that 85% of all biomedical research is 'wasted', and that a significant proportion of this can be attributed to problems choosing and reporting relevant outcomes in trials.[5] Thus, if we combine the number of patients who have participated in gastric cancer surgery trials over the last two decades with those that actively recruiting trials wish to attract, a total of 20 000 patients may have participated in trials which, from a methodological perspective, could have had a far greater benefit and impact. Not only does this represent an inefficient use of time and scarce financial resource, but it may also have a longer-lasting negative impact on future trial participation by patients. Similar issues related to outcome reporting have been identified by several other groups supporting the theory that this is a widespread problem.[6–8 11]

Furthermore, if the methodology of a particular trial is not sufficiently robust or the outcomes reported are not relevant to key stakeholders, the natural course will be for other researchers to examine the same interventions again, using a different approach. If these subsequent trials do not address the underlying methodological issues, they only contribute to a perpetual cycle which serves to weaken the evidence base. This is reflected within the field of gastric cancer surgery where 13 trials have examined minimally invasive gastrectomy and a further 13 are actively recruiting to trials examining the same intervention.

Reported outcomes should be relevant to key stakeholders including patients. Given the sheer volume of complications that are reported by gastric surgical trials, one may expect to find the impact on QoL is routinely reported. Indeed, QoL has been demonstrated as an important outcome to measure in other gastrointestinal cancer fields.[37] This has certainly not been the case with gastric cancer surgery trials over the last two decades and while there seems to be a greater acceptance by trials currently in recruitment that QoL is important to measure (although this group still represents less than half of ongoing trials), there remains great variation in relation to 'how' and 'when' it is measured.

To address these inconsistencies, we believe that a COS is required for gastric cancer surgery studies. Developing a minimum reporting standard will contribute to maximising the benefits from randomised control trials which are expensive, labour intensive and logistically challenging to set up. A COS does not aim to restrict the outcomes that are reported, but merely to ensure that the most critical outcomes (as decided by key stakeholders) are clearly defined and measured uniformly.

The challenges associated with inconsistent outcome reporting in trials are certainly not confined to the field of gastric cancer. The COMET initiative database (http://www.comet-initiative.org/studies/search) contains details of >400 completed, active or planned COS projects from across many different specialities.[38] While experience within this relatively new research field has grown considerably over the last decade, there is still much work to be done to further develop the various methodological

approaches which can be applied. The GASTROS study aims to add to this in several ways including examining the role of 'internationalising' COS development by undertaking a multilanguage Delphi survey as part of a consensus-seeking process.

## Strengths and limitations

In addition to being the first systematic review to examine this subject, this study is based on a reproducible and transparent methodology which has been subjected to critical appraisal from a study management team and peer-review process; a protocol of the GASTROS study which aims to develop this COS has been published previously.[4] Nonetheless, there are limitations. Including non-English and non-randomised studies in our search strategy may have identified other different outcomes reported in this field. However, when finalising our inclusion criteria for this review, the two primary objectives of this review were considered: (1) to describe the current landscape of outcome reporting in gastric cancer surgery RCTs and (2) to take forward a 'long list' of outcomes to be prioritised (by means of a Delphi survey) to form the basis of a COS for RCTs. While we accept that such a COS would have benefits to non-RCTs and national audits, our primary focus was to improve the quality of RCTs and hence excluding other study types. In addition, there will be an opportunity during the Delphi survey (stage 2 of the GASTROS study) for participants to add further outcomes (not already identified from this review) which key stakeholders deem important to be considered for prioritisation. A further limitation to this review was that it was not prospectively registered on a public database. However, as we describe above, the GASTROS study, including its scope and systematic review plan, has been peer-reviewed and published previously.[4]

In summary, the reporting of outcomes in gastric cancer surgery trials is inconsistent and there is large variation with respect to definitions, measurement tools and timing of measurement. This means that data cannot be synthesised efficiently. We believe that a COS to define a minimum set of standards to implement across all gastric surgical trial is warranted.

**Author affiliations**
[1]Department of Oesophago-Gastric Surgery, Manchester Royal Infirmary, Manchester University NHS Foundation Trust, Manchester Academic Health Science Centre, Manchester, UK
[2]Department of Oesophago-Gastric Surgery, Salford Royal Hospital, Salford Royal NHS Foundation Trust, Manchester Academic Health Science Centre, Manchester, UK
[3]Division of Cancer Sciences, School of Medical Sciences, Faculty of Biology, Medicine and Health, University of Manchester, Manchester, UK
[4]Centre for Surgical Research, University of Bristol, Bristol, UK
[5]National Institute for Health Research, Bristol Biomedical Research Centre, Bristol, UK
[6]MRC North West Hub for Trials Methodology Research, University of Liverpool, Liverpool, UK
[7]Paediatric ENT Department, Royal Manchester Children's Hospital, Manchester University NHS Foundation Trust, Manchester Academic Health Science Centre, Manchester, UK

[8]Division of Infection, Immunity and Respiratory Medicine, School of Biological Sciences, Faculty of Biology, Medicine and Health, University of Manchester, Manchester, UK

[9]Division of Dentistry, School of Medical Sciences, Faculty of Biology, Medicine and Health, University of Manchester, Manchester, UK

**Acknowledgements** The authors thank the patient, nurse and clinician representatives on the GASTROS Study Advisory Group for their continued advice and invaluable contribution to this study.

**Contributors** BA contributed to the study design and drafted the m anuscript. BA and A-MG extracted the data from manuscripts. A-MG, JMB, PRW and IAB contributed to the study design and revised the manuscript. All authors read and approved the final manuscript.

**Funding** This study was funded by the National Institute for Health Research (NIHR) Doctoral Research Fellowship Grant programme (DRF-2015-08-023).

**Competing interests** None declared.

**Patient consent** Not required.

**Provenance and peer review** Not commissioned; externally peer reviewed.

**Data sharing statement** Upon the completion of the study, the original data file (consisting of extracted data from trial papers) is available upon request from the corresponding author, following discussion and agreements by the study management team.

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
