## [Reviewer comments · BMJ Open]

ARTICLE DETAILS

TITLE (PROVISIONAL)	Reporting of Outcomes in Gastric Cancer Surgery Trials: A Systematic Review.
AUTHORS	Alkhaffaf, Bilal; Blazeby, Jane; Williamson, Paula; Bruce, Iain; Glenny, Anne-Marie

VERSION 1 – REVIEW

REVIEWER	Mitsuru Sasako Hyogo College of Medicine Japan
REVIEW RETURNED	07-Feb-2018

GENERAL COMMENTS	The authors report results of systematic review of gastric cancer studies comparing two surgical procedures. 1. They included many studies of randomized phase-II studies and single and single institutional studies. Quality of studies in multi-center phase-III studies seems different from the formers. Inclusion of the former studies give a wrong impression of quality of clinical studies including how to report results. They should exclude the former studies or at least separately analyze them from high quality studies.2. Importance of QOL issues is much higher in non-inferiority studies than superiority studies. These two subgroups are also separately analysed to get more practical conclusions.3. There remains some surgeons who don't understand the aim of their RCT. They don't discriminate randomizes phase-II and confirmatory phase-III. For different aims of these two categories, requested or important outcomes maybe different.4. They should neglect all studies in which sample size calculations based on the primary endpoint are not clearly written.5. In their arguments, they don't refer to quality of surgery in RCTs and quality control in multicenter studies. Even in a single institutional study, it is very important to describe the detail of surgery which may affect results of the study. Otherwise, applicability of the results is doubtful.6. Unifying the way to report outcome in several aspects, such as morbidity or QOL, is of course quite important but how to set up surgical studies is more important if they want to improve the quality of RCTs. Definition of each surgical procedure and quality control of treatment itself is unique to surgical trials and therefore needed to be established.
--

REVIEWER	L. Huang Division of Clinical Epidemiology and Aging Research, German Cancer Research Center; Medical Faculty Heidelberg; Department of Gastrointestinal Surgery, First Affiliated Hospital of Anhui Medical University
-----------------	--

REVIEW RETURNED	12-Feb-2018
-------------

GENERAL COMMENTS	This paper investigated outcome reporting in gastric cancer surgery trials, highlighting great inconsistencies and variations. Abstract: -“A total of 749 outcomes were reported of 96 (13 per cent) were accompanied by an attempted definition.” This sentence seems problematic. Strengths & Limitations of the Study: -The limitations seem not to have been depicted. Main text: -“No single outcome was reported in every trial.” Each trial might have its specific own aims and focus (e.g., short-term perioperative outcomes including feasibility and safety, long-term oncologic and survival outcomes, and quality of life). Thus it should not be regarded as a surprising finding herein. The authors are strongly encouraged to have a more specific focus or to conduct subgroup analysis according to the timeline of the reportable outcomes in surgical investigations. -Indeed, perioperative mortality should be standardized in definition. Both 30-day mortality and mortality during hospitalization are important aspects to report during the short term. For the longer term, it is suggested to report the total mortality along with the survival at specific time points, as mortality could reveal some aspects not covered by the survival measure. -For adverse events, indeed, they are observed via clinical and/or radiological examinations. While it is important to have common standards for definition, it is also important to encourage novel assessment of modified protocols which might be able to more reasonably categorize the complications. Furthermore, geographic disparities might call for local adjustment of a specific standard. -Results of reporting on surgical outcomes (e.g., surgical time, analgesic use, and estimated blood loss) need to be reported by the authors. -Regarding survival outcomes, the authors are encouraged to further describe their findings on disease-free survival, relapse-free survival, progression-free survival, and cancer-specific survival, et al., and to make their corresponding comments. -Overall, I do believe that the variation in outcome reporting could add to the comprehensiveness as well in literature regarding surgical trials, since it is impossible for a single trial to cover all the aspects. Diversity with discrepant focuses should not be discouraged, as long as there is no scientific flaws imbedded. Nevertheless, a recommendation list proposing the priority of reporting of measurements for a trial with a specific aim is encouraged from the authors.
---

REVIEWER	Luca Bertolaccini Maggiore Teaching Hospital, Bologna (IT)
REVIEW RETURNED	23-Feb-2018

GENERAL COMMENTS	I have the following questions for you, which I believe, need to be addressed before publication: First, please add if you perform a librarian certified search and add the criteria in the supplemental file. Secondly, please add if you registered the meta-analysis in the PROSPERO International prospective register of systematic reviews (https://www.crd.york.ac.uk/PROSPERO/). The statistical analysis should be rewritten according to one of the
--

	recently published guidelines (e.g. Hickey GL, Dunning J, Seifert B, Sodeck G, Carr MJ, Beyersdorf F on behalf of the EJCTS and ICVTS Editorial Committees Editor's Choice: Statistical and data reporting guidelines for the European Journal of Cardio-Thoracic Surgery and the Interactive CardioVascular and Thoracic Surgery. Eur J Cardiothorac Surg 2015;48:180-93). In the results, please add some meta-analysis of the data reported in the tables. The discussion should be improved with a better literature search showing the differences of this study from the others recent meta-analyses. About minor points, there are typos and grammars errors in the text. Please thoroughly check the article. References should be reported according to the authors' instructions. The paper should be formatted according to the authors' instructions.
--	---

REVIEWER	Sam Adie St. George and Sutherland Clinical School University of New South Wales Sydney, Australia
REVIEW RETURNED	22-Mar-2018

GENERAL COMMENTS	Introduction - I appreciate this paper is part of a wider study (GASTROS), but can the authors clearly differentiate the aims of this paper from the wider aims of other components of GASTROS? At the moment this is only dealt with in a single line "This review specifically aims to examine the degree of variation in the reporting of outcomes described by gastric cancer surgery trials" which seems a little inadequate for an aims statement. Methods - What was the rationale for only including Type 2 studies? Why weren't surgical vs. non-surgical trials also included? Timeline - I appreciate a cutoff has to be established for how far back trials are searched, but what was the rationale for using the publication of the first CONSORT statement? Was a time delay needed so that the recommendations of CONSORT were disseminated? Demographics - "demography" used in this context is somewhat confusing, as it implies data regarded to a population of persons, rather than included trials. Perhaps this can be changed to "Trial characteristics". Outcomes - It may be useful here to define what an outcome was. In my experience some trials may report data in a single arm of the study, with no relative comparison. This data may not be regarded as an outcome according to widely accepted definitions e.g. from the U.S. NIH.
--

Rationalising and grouping outcomes

- Can you clarify what you mean by “merged” outcomes here? Do you mean an outcome that was spelled or written differently in a single report of a trial, or was it in groups of trials?

- “Outcomes were organized into ‘outcome themes’...”. Given this appears to be the main focus of the study (and more broadly that GASTROS aims to improve the methods in gastric surgery trials), I would expect much more detail here. How did this process of organizing themes take place? The process appears to involve a classification of a large set of outcomes into groups, which has inherent issues with validity. Were the “broad categories” determined a priori? How were these categories generated? How many authors took part in the classification? Was the validity of the classification tested? In the following section there is mention of a “data extraction process” and a “study management group” where agreement was assessed. Can details regarding this process be provided here in the methods section? As a rule of thumb, it would be preferable to have enough detail for replication of the study, or so that researchers in other fields can replicate the methods if GASTROS is ultimately successful in achieving its aims.

Patient and public involvement

- Its great that the advisory group included patients. Can you provide a little more detail into what was actually contributed for this particular paper by the stakeholders?

Results

- This is generally well written and easy to follow. I note that only specific outcome set subheadings were used however. Why weren't the other outcome sets discussed? Word limits?

Discussion

- This is well written and easy to follow. It also sets out the next stage of the overall project.

- Can the authors please comments regarding the validity of the classification as mentioned above.

- Can the authors also provide any examples in other clinical areas where similar efforts took place to standardize an outcome set. E.g. the OMERACT initiative, diabetes trials?

Figure 1

- Can you clarify why 31 articles were excluded after “full text screen” and a further 20 articles were excluded “during data extraction”

Figure 2

- Can you rephrase the title of this figure. “proposed outcome themes to be reported in future trials” is a bit confusing and may imply that this is what the authors propose (in line with the broad aims of GASTROS mentioned in the introduction).

Table 3

	- This table is very interesting. In particular, I find it quite shocking that only 19% of trials contained patient reported measures, and less than half reported mortality! - I suggest the authors divide the mortality theme into two distinct themes- short term mortality and long term mortality, since these outcomes have radically different implications for a treatment. Short term mortality tends to be a measure of harm, while long term mortality is a measure of efficacy. I appreciate that the authors have discussed them separately in the body of the manuscript, but they should also be clearly divided in the main results table to prevent conflation. -It would be useful and interesting (although not mandatory) if the information in this table is presented in a diagram form, such as a Venn diagram. Overlapping circles can be used to represent outcome themes. PRISMA checklist - I appreciate that some PRISMA items are not applicable to this study, but I would argue items 10, 11, and 13 require more information in the “rationalizing and grouping outcomes” section as stated above. - was the review registered / is this information available in the previously cited protocol in Trials?
--	---

VERSION 1 – AUTHOR RESPONSE

Reviewer(s)' Comments to Author:

Reviewer: 1

Reviewer Name: Mitsuru Sasako

Institution and Country: Hyogo College of Medicine, Japan

Please state any competing interests or state 'None declared': None

Please leave your comments for the authors below. The authors report results of systematic review of gastric cancer studies comparing two surgical procedures.

1. They included many studies of randomized phase-II studies and single and singleinstitutional studies. Quality of studies in multi-center phase-III studies seems different from the formers. Inclusion of the former studies give a wrong impression of quality of clinical studies including how to report results. They should exclude the former studies or at least separately analyze them from high quality studies.

ACTION: We have singled out multi-centre trials for a separate analysis in the results section. This however still demonstrates large variations in this group.

2. Importance of QOL issues is much higher in non-inferiority studies than superiority studies. These two subgroups are also separately analysed to get more practical conclusions.

RESPONSE: It is interesting that QOL may be considered to be a more important outcome in non-inferiority studies than superiority (assuming that the primary outcome is survival), however, in this review we are not examining how results are used in decision-making. The purpose of the GASTROS study is to determine *which* outcomes are important to key

stakeholders, including researchers and particularly patients. Based on this review, we have demonstrated that assumptions previously made by trialists as to which outcomes should be measured, results in significant heterogeneity.

3. There remains some surgeons who don't understand the aim of their RCT. They don't discriminate randomizes phase-II and confirmatory phase-III. For different aims of these two categories, requested or important outcomes maybe different.

RESPONSE: We agree with the reviewer that there is much work to be done in educating surgical researchers and improving the quality of surgical trials. Indeed the authors have been working hard over the past decade to educate surgeons to do more and better trials:

Delivering successful randomized controlled trials in surgery: Methods to optimize collaboration and study design. Blencowe NS, Cook JA, Pinkney T, Rogers C, Reeves BC, Blazeby JM. Clin Trials. 2017 Apr;14(2):211-218. doi: 10.1177/1740774516687272. Epub 2017 Jan 31.

However, the purpose of the review was two-fold; firstly, to demonstrate heterogeneity in outcome reporting, and secondly to identify outcomes to be prioritised in a future international survey. As such, we are keen to include outcomes from all phase RCTs in this review. We have addressed differences between single institutional and multi-centre trials as requested above. Ultimately, the merit and relevance of individual outcomes will be prioritised by the participants.

4. They should neglect all studies in which sample size calculations based on the primary endpoint are not clearly written.

RESPONSE: We thank the reviewer for pointing out that many surgical trials are not well designed or reported. We agree that where no calculations are based on the primary endpoint it reflects concerns about study quality. In this review, however, we are concerned about the choice of outcome, not the actual effect size. We have therefore decided to retain such studies.

5. In their arguments, they don't refer to quality of surgery in RCTs and quality control in multicenter studies. Even in a single institutional study, it is very important to describe the detail of surgery which may affect results of the study. Otherwise, applicability of the results is doubtful.

RESPONSE: We agree with the reviewer that the reporting of quality assurance for surgical techniques is important to determine to external validity of the results. However, the scope of the GASTROS study and, this review, focuses purely on outcome reporting in this field. We have included a reference to a similar publication from the surgical field to demonstrate the relevance of such work which informed the development of a core outcome set.

Blencowe NS, Strong S, McNair AG, Brookes ST, Crosby T, Griffin SM, Blazeby JM. Reporting of short-term clinical outcomes after esophagectomy: a systematic review. Ann Surg. 2012 Apr;255(4):658-66.

6. Unifying the way to report outcome in several aspects, such as morbidity or QOL, is of course quite important but how to set up surgical studies is more important if they want to improve the quality of RCTs. Definition of each surgical procedure and quality control of treatment itself is unique to surgical trials and therefore needed to be established.

RESPONSE: We agree with the reviewer that there is much work which is required to improve the quality of surgical trials. The reporting of outcomes is just one aspect which the GASTROS study aims to address. We would support and encourage an initiative to address the standardisation of surgical techniques as the reviewer has pointed out, however this is outside the scope of our study.

Reviewer: 2

Reviewer Name: L. Huang

Institution and Country: Division of Clinical Epidemiology and Aging Research, German Cancer Research Center; Medical Faculty Heidelberg; Department of Gastrointestinal Surgery, First Affiliated Hospital of Anhui Medical University

Please state any competing interests or state 'None declared': None declared.

Please leave your comments for the authors below This paper investigated outcome reporting in gastric cancer surgery trials, highlighting great inconsistencies and variations.

Abstract:

-“A total of 749 outcomes were reported of 96 (13 per cent) were accompanied by an attempted definition.” This sentence seems problematic.

ACTION: We have missed the word ‘which’ before the figure 96. This has been corrected.

Strengths & Limitations of the Study:

-The limitations seem not to have been depicted.

ACTION: We have revised this section to more clearly describe the strengths/limitations of the study in the style of recent systematic review publications from BMJ Open.

Main text:

-“No single outcome was reported in every trial.” Each trial might have its specific own aims and focus (e.g., short-term perioperative outcomes including feasibility and safety, long-term oncologic and survival outcomes, and quality of life). Thus it should not be regarded as a surprising finding herein. The authors are strongly encouraged to have a more specific focus or to conduct subgroup analysis according to the timeline of the reportable outcomes in surgical investigations.

RESPONSE: The premise of the GASTROS study is to demonstrate and address the variation in outcome reporting in surgical effectiveness trials and not feasibility and safety studies. Trials are resource intensive and should help clinicians and patients understand which treatments are best; this can be done through evidence synthesis which cannot be performed if trials measure outcomes differently. Whilst we do not suggest that all trials need to have the same primary outcomes, there should be a group of critically important outcomes (important to researchers and patients) which should be reported and measured by all trials in the same way to address this issue. Grouping of outcomes into early, intermediate, late categories is not part of the current COS development methodology, which is focused on *what* are the most important outcomes to measure in effectiveness trials (please visit www.comet-initiative.org for more information).

-Indeed, perioperative mortality should be standardized in definition. Both 30-day mortality and mortality during hospitalization are important aspects to report during the short term. For the longer term, it is suggested to report the total mortality along with the survival at specific time points, as mortality could reveal some aspects not covered by the survival measure.

RESPONSE: We agree that the reporting of outcomes should be standardised; *how* outcomes should be measured will be addressed in future stages of the GASTROS study. We would refer you to our published protocol for further details of our planned work:

Alkhaffaf B, Glenny AM, Blazeby JM, Williamson P, Bruce IA. Standardising the reporting of outcomes in gastric cancer surgery trials: protocol for the development of a core outcome set and accompanying outcome measurement instrument set (the GASTROS study). *Trials*. 2017 Aug 9;18(1):370. doi: 10.1186/s13063-017-2100-7.

-For adverse events, indeed, they are observed via clinical and/or radiological examinations. While it is important to have common standards for definition, it is also important to encourage novel assessment of modified protocols which might be able to more reasonably categorize the complications. Furthermore, geographic disparities might call for local adjustment of a specific standard.

RESPONSE: We agree with the reviewer. The methods of measuring outcomes will be dealt with at a future stage of the GASTROS study which has been previously described in our published protocol.

-Results of reporting on surgical outcomes (e.g., surgical time, analgesic use, and estimated blood loss) need to be reported by the authors.

RESPONSE: As described above, the aim of the study is not to report the results of studies included within the analysis, but to demonstrate what, when and how outcomes were reported. We have included all outcomes which were reported within the supplementary appendix. We believe that the outcome themes described in the results section comprehensively demonstrate the variation in outcome reporting within this field without the need for further analysis.

-Regarding survival outcomes, the authors are encouraged to further describe their findings on disease-free survival, relapse-free survival, progression-free survival, and cancer-specific survival, et al., and to make their corresponding comments.

RESPONSE: The scope of this study is not to report the *actual* outcomes from these trials, but to demonstrate the variation in the reporting of these outcomes. We have summarised the reporting of survival outcomes in table 4.

-Overall, I do believe that the variation in outcome reporting could add to the comprehensiveness as well in literature regarding surgical trials, since it is impossible for a single trial to cover all the aspects. Diversity with discrepant focuses should not be discouraged, as long as there is no scientific flaws imbedded. Nevertheless, a recommendation list proposing the priority of reporting of measurements for a trial with a specific aim is encouraged from the authors.

RESPONSE: We absolutely agree that a MINIMUM standard of reporting is important and that the measurement of other outcomes would be expected.

Reviewer: 3

Reviewer Name: LucaBertolaccini

Institution and Country: Maggiore Teaching Hospital, Bologna (IT)

Please state any competing interests or state 'None declared': None declared

Please leave your comments for the authors below Thank you for submitting this article to the BMJ Open. I was pleased to receive it as a reviewer.

It is indeed an exciting paper since you aim to analysis the survival time and recurrence rate of limited resection and lobectomy with and without employ propensity score matching studies.

I have the following questions for you, which I believe, need to be addressed before publication: First, please add if you perform a librarian certified search and add the criteria in the supplemental file.

Secondly, please add if you registered the meta-analysis in the PROSPERO International prospective register of systematic reviews (<https://urldefense.proofpoint.com/v2/url?u=https-3A-www.crd.york.ac.uk-PROSPERO-&d=DwIFaQ&c=bMxC-A1upgdsx4J2OmDkk2Eep4PyO1BA6pjHrrW-ii0&r=cCBH3peyAhqeqZPMNXa51UxytPdYs4KIO7x->

XIkFLRk&m=W7gw02il6hv-RQAMAs4PtKvs6mOQk-XpwN_mwvQgK-c&s=MR4vJL83mbrvpzgtzYODS6HRn3ybveOa2Rpv7AcOfM&e=).

The statistical analysis should be rewritten according to one of the recently published guidelines (e.g. Hickey GL, Dunning J, Seifert B, Sodeck G, Carr MJ, Beyersdorf F on behalf of the EJCTS and ICVTS Editorial Committees Editor's Choice: Statistical and data reporting guidelines for the European Journal of Cardio-Thoracic Surgery and the Interactive CardioVascular and Thoracic Surgery. Eur J Cardiothorac Surg 2015;48:180-93).

In the results, please add some meta-analysis of the data reported in the tables.

The discussion should be improved with a better literature search showing the differences of this study from the others recent meta-analyses.

About minor points, there are typos and grammars errors in the text. Please thoroughly check the article.

References should be reported according to the authors' instructions.

The paper should be formatted according to the authors' instructions.

Good luck with your article, and thanks again for submitting it.

RESPONSE: We believe that this review has been included in error as it refers to a different paper and topic.

Reviewer: 4

Reviewer Name: Sam Adie

Institution and Country: St. George and Sutherland Clinical School, University of New South Wales, Sydney, Australia

Please state any competing interests or state 'None declared': None declared

Please leave your comments for the authors below

Thank you for the opportunity to review this paper.

The authors evaluated the variation in outcome reporting in recently published trials of gastric surgery that compared a surgical intervention to another surgical intervention. The general issue of outcome reporting is topical and important, although limited to a specific subspecialty condition.

Comments-

Introduction

- I appreciate this paper is part of a wider study (GASTROS), but can the authors clearly differentiate the aims of this paper from the wider aims of other components of GASTROS? At the moment this is only dealt with in a single line "This review specifically aims to examine the degree of variation in the reporting of outcomes described by gastric cancer surgery trials" which seems a little inadequate for an aims statement.

ACTION: We have elaborated further in the introduction section as to the basis of the study and why the systematic review is important and relevant.

Methods

- What was the rationale for only including Type 2 studies? Why weren't surgical vs. non-surgical trials also included?

RESPONSE: The rationale for this has been described in detail within our previously published protocol (referenced below this response) but was not included in detail due to brevity; we have elaborated in the text our justification for this. The volume of active

and future planned Type 2 RCTs is such that the development of a purely surgical core outcome set is warranted. We recognise however that the development of a core outcome set to include other treatment modalities is important and may be part of future work by our group. Future related COS in this area may allow for comparison between surgical, minimally invasive and non-surgical modalities, but the volume of work would not be realistic to complete in a single project, and there would be a risk of the generated COS becoming too generalised if the selected outcomes had to be relevant to all modalities.

Alkhaffaf B, Glenny AM, Blazeby JM, Williamson P, Bruce IA. Standardising the reporting of outcomes in gastric cancer surgery trials: protocol for the development of a core outcome set and accompanying outcome measurement instrument set (the GASTROS study). *Trials*. 2017 Aug 9;18(1):370. doi: 10.1186/s13063-017-2100-7.

Timeline

- I appreciate a cutoff has to be established for how far back trials are searched, but what was the rationale for using the publication of the first CONSORT statement? Was a time delay needed so that the recommendations of CONSORT were disseminated?

RESPONSE: We appreciate that a time delay may have been required for the dissemination of the CONSORT recommendations, however, it would be difficult to accurately assume how long this should be. As the reviewer appreciates, a cut-off was required, but in addition, we wanted to ensure 20 years of trials to comprehensively identify outcomes for use in later stages of the GASTROS study.

Demographics

- “demography” used in this context is somewhat confusing, as it implies data regarded to a population of persons, rather than included trials. Perhaps this can be changed to “Trial characteristics”.

ACTION: This has been changed.

Outcomes

- It may be useful here to define what an outcome was. In my experience some trials may report data in a single arm of the study, with no relative comparison. This data may not be regarded as an outcome according to widely accepted definitions e.g. from the U.S. NIH.

ACTION: This has been detailed in our published protocol (referenced above). We have included the definition now within the text.

Rationalising and grouping outcomes

- Can you clarify what you mean by “merged” outcomes here? Do you mean an outcome that was spelled or written differently in a single report of a trial, or was it in groups of trials?

RESPONSE: All outcomes were transcribed verbatim. In this instance, merging of outcomes refers only to ‘cleaning up’ variations of spelling (e.g. plural and singular versions – we use anastomotic leak/leaks and leakages as an example). This occurred when there were different spellings within the same publication, within different publications within the same trial and with different publications from different trials. We have clarified this within the manuscript.

- “Outcomes were organized into ‘outcome themes’...”. PubMed Given this appears to be the main focus of the study (and more broadly that GASTROS aims to improve the methods in gastric surgery trials), I would expect much more detail here. How did this process of organizing themes take place? The process appears to involve a classification of a large set of outcomes into groups, which has

inherent issues with validity. Were the “broad categories” determined a priori? How were these categories generated? How many authors took part in the classification? Was the validity of the classification tested? In the following section there is mention of a “data extraction process” and a “study management group” where agreement was assessed. Can details regarding this process be provided here in the methods section? As a rule of thumb, it would be preferable to have enough detail for replication of the study, or so that researchers in other fields can replicate the methods if GASTROS is ultimately successful in achieving its aims.

RESPONSE: Outcomes were organised according to categories used by similar groups developing core outcome sets within the field of surgery. We have now included more detail in the methods section with respect to this valid point and referenced a new proposed taxonomy in the field of outcome reporting (referenced below).

Dodd S, Clarke M, Becker L, Mavergames C, Fish R, Williamson PR. A taxonomy has been developed for outcomes in medical research to help improve knowledge discovery. J Clin Epidemiol. 2018 Apr;96:84-92. doi: 10.1016/j.jclinepi.2017.12.020. PubMed Epub 2017 Dec 28.

There were no disagreements between the two reviewers that required referral to the study management group. The text has been updated to reflect this.

Patient and public involvement

- Its great that the advisory group included patients. Can you provide a little more detail into what was actually contributed for this particular paper by the stakeholders?

ACTION: The review results were presented to the Study Advisory Group. The ensuing discussion emphasised the importance of PROs which was highlighted within the results section. This contribution has been added to the methods section.

Results

- This is generally well written and easy to follow. I note that only specific outcome set subheadings were used however. Why weren't the other outcome sets discussed? Word limits?

ACTION: The reviewer is correct in that there had to be a cut off. The full list of outcomes reported in trials is available in a supplementary appendix. We believe that the results presented achieve the aims of the study which is to demonstrate the degree of variation in outcome reporting.

Discussion

- This is well written and easy to follow. It also sets out the next stage of the overall project.
- Can the authors please comments regarding the validity of the classification as mentioned above.

RESPONSE: The manuscript has now been amended to address this point within the methods section. Whilst we acknowledge that this is not a validated classification system, we opted to use a system used by a) key researchers within the field of outcomes research and b) by other groups developing COS in gastro-intestinal surgery. We have updated the manuscript to reflect our justification for this.

- Can the authors also provide any examples in other clinical areas where similar efforts took place to standardize an outcome set. E.g. the OMERACT initiative, diabetes trials?

ACTION: We have discussed the large number of projects which are attempting to address the challenges of heterogeneity in outcome reporting, including reference to over 400 COS studies details on the COMET database. We have added a link in the discussion section so that readers can examine the areas in which these projects are taking place.

Figure 1

- Can you clarify why 31 articles were excluded after “full text screen” and a further 20 articles were excluded “during data extraction”

ACTION: We thank the reviewer for noting this error in our manuscript. 40 of the systematic reviews were excluded as this was a review of RCTs. The SRs were included to identify any further RCTs (this has been added to the text); no further RCTs were identified. This has now been corrected and the flow chart has been updated with reasons for exclusions.

Figure 2

- Can you rephrase the title of this figure. “proposed outcome themes to be reported in future trials” is a bit confusing and may imply that this is what the authors propose (in line with the broad aims of GASTROS mentioned in the introduction).

ACTION: The entire title has been changed to ‘Outcome themes reported in a) gastric cancer surgery trials and b) in future trials based on study protocols.’

Table 3

- This table is very interesting. In particular, I find it quite shocking that only 19% of trials contained patient-reported measures, and less than half reported mortality!
- I suggest the authors divide the mortality theme into two distinct themes- short term mortality and long term mortality, since these outcomes have radically different implications for a treatment. Short term mortality tends to be a measure of harm, while long term mortality is a measure of efficacy. I appreciate that the authors have discussed them separately in the body of the manuscript, but they should also be clearly divided in the main results table to prevent conflation.
-It would be useful and interesting (although not mandatory) if the information in this table is presented in a diagram form, such as a Venn diagram. Overlapping circles can be used to represent outcome themes.

ACTION: The mortality theme has been grouped together in the same way that other surgical COS groups have (described above and referenced in the manuscript). The short and long-term mortality outcomes have been described in the text of the results section. We have split into long/short term mortality in the table as requested.

PRISMA checklist

- I appreciate that some PRISMA items are not applicable to this study, but I would argue items 10, 11, and 13 require more information in the “rationalizing and grouping outcomes” section as stated above.

ACTION: We have addressed these issues in the text as referred to in previous answers. ‘Summary measures’ were not applicable in this study.

- was the review registered / is this information available in the previously cited protocol in Trials?

RESPONSE: The review was not registered; this was an oversight and we were not able to retrospectively register the review on PROSPERO. Nonetheless, details of the systematic review were included in the published protocol and the process was subject to scrutiny by the study management and study advisory group. In addition, the entire study, including the systematic review plan, was peer-reviewed by a national governmental grant-awarding body.

VERSION 2 – REVIEW

REVIEWER	Lei Huang German Cancer Research Center (DKFZ), Germany; First Affiliated Hospital of Anhui Medical University, China
REVIEW RETURNED	01-Jun-2018

GENERAL COMMENTS	My comments have been appropriately addressed.
--

REVIEWER	Sam Adie St. George and Sutherland Clinical School, University of New South Wales, Sydney, Australia
REVIEW RETURNED	20-Jun-2018

GENERAL COMMENTS	The authors have addressed the major comments adequately, including clarifying their objectives, providing a rationale for their outcome themes, and re-classifying mortality as an outcome. The manuscript is stronger as a result. Minor comments: “Doing enables COS researchers....” Is a typo. The authors state “There were no disagreements between the two reviewers that required referral to the study management group. The text has been updated to reflect this.” Can you clarify whether this means you had identical answers to each other? What was the level of disagreement in the initial classification? In the results section, the authors have only discussed a subset of outcome themes, with the rest in Appendix 2. Can the authors make it a bit more explicit that only the most important themes are presented in the results, with the rest of the information in Appendix 2? I am not sure if Table 3 really adds anything to the actual body of the paper- I will leave this up to the editors. It should be included as an Appendix.
---

VERSION 2 – AUTHOR RESPONSE

1. “Doing enables COS researchers....” Is a typo.

This has been amended to ‘Doing so enables COS researchers...’

2. The authors state “There were no disagreements between the two reviewers that required referral to the study management group. The text has been updated to reflect this.” Can you clarify whether this means you had identical answers to each other? What was the level of disagreement in the initial classification?

During the initial data extraction process, both reviewers extracted outcomes from the same group of 10 publications. The data was compared between them. In the first batch of ten

publications, there were some minor clarifications sought by one the reviewers (AMG), but the outcomes extracted by both reviewers were otherwise identical.

The manuscript was previously updated to 'there were no *unresolved* disagreements that required referral to the GASTROS study management team for a final decision'.

- 3. In the results section, the authors have only discussed a subset of outcome themes, with the rest in Appendix 2. Can the authors make it a bit more explicit that only the most important themes are presented in the results, with the rest of the information in Appendix 2?**

We have updated the manuscript with the sentence 'Below, we present a summary of some of the most commonly reported short and long-term outcome themes.' We prefer not to use the term 'most important' because we do not know if the themes and outcomes reported are 'most important' to all key stakeholders in this research field.